

# Anticipatory Postural Adjustments and kinematic arm features when postural stability is manipulated

Bianca Callegari[1], Ghislain Saunier[2], Manuela Brito Duarte[1],
Gizele Cristina da Silva Almeida[1], Cesar Ferreira Amorim[3], France Mourey[4],
Thierry Pozzo[4,5] and Givago da Silva Souza[6]

[1] Laboratório de Estudos da Motricidade Humana, Universidade Federal do Pará, Belém, Brazil
[2] Laboratorio de Cognição Motora, Universidade Federal do Pará, Belém, Brazil
[3] Doctoral and Masters Program in Physical Therapy, Universidade Cidade de São Paulo, São Paulo, Brazil
[4] INSERM U1093, Cognition Action et Plasticité Sensori-motrice, Université de Bourgogne, Dijon, France
[5] Italian Institute of Technology CTNSC@UniFe (Center of Translational Neurophysiology for Speech and Communication), Italian Institute of Technology, Ferrara, Italy
[6] Núcleo de Medicina Tropical, Instituto de Ciencias Biológicas, Universidade Federal do Pará, Belém, Brazil

Corresponding author
Bianca Callegari, callegari@ufpa.br, callegaribi@uol.com.br

## ABSTRACT

Beyond the classical paradigm that presents the Anticipatory Postural Adjustments (APAs) as a manner to create forces that counteract disturbances arising from the moving segment during a pointing task, there is a controversial discussion about the role APAs to facilitate the movement and perform a task accurately. In addition, arm kinematics features are classically used to infer the content of motor planning for the execution and the control of arm movements. The present study aimed to disentangle the conflicting role of APAs during an arm-pointing task in which the subjects reach a central diode that suddenly turns on, while their postural stability was manipulated. Three postures were applied: Standing (Up), Sit without feet support (SitUnsup) and Sit with feet support (SitSup). We found that challenging postural stability induced an increase of the reaction time and movement duration (observed for the SitUnsup compared to SitSUp and Up) as well as modified the upper-limb velocity profile. Indeed, a greater max velocity and a shorter deceleration time were observed under the highest stability (SitSup). Thus, these Kinematics features reflect less challenging task and simple motor plan when the body is stabilized. Concerning the APAs, we observed the presence of them independently of the postural stability. Such a result strongly suggests that APAs act to facilitate the limb movement and to counteract perturbation forces. In conclusion, the degree of stability seems particularly tuned to the motor planning of the upper-limb during a pointing task whereas the postural chain (sitting vs. standing) was also determinant for APAs.

## INTRODUCTION

Postural perturbations induced by fast arm movements in a standing condition were widely studied and generally displace the body's center of mass (COM) causing a disruption of

posture. Classically, it is accepted that anticipatory postural adjustments (APAs) begin before the limb upward movement taking place to counteract the expected mechanical effects of the focal perturbation in a feedforward way (*Moore et al., 1992*; *Santos, Kanekar & Aruin, 2010*). Authors suggest that they are generated at a high level of the central nervous system (CNS) promoting earlier changes in the activity of the postural muscles in order to compensate for a potential shift in the COM (*Sijper & Latash, 2000*; *Yoshida et al., 2008*).

Beyond this classical paradigm that presents APAs as a manner to create forces that counteract disturbances arising from the moving segment, there is a controversial discussion about the role of present APAs. For instance, *Stapley et al. (1999)* found that, rather than acting to stabilize the COM, APAs created necessary conditions for forward COM displacement within the base of support in the upright position. *Tijtgat et al. (2013)* investigated APA in one-handed ball catching and proposed that APAs besides a stabilization of some body part are functionally link to the task specificity and ensure an active role in task goal achievement.

Their results challenge the classical role assigned to APAs in the literature. In this paper we propose to contribute to this controversial issue by studying APA in a stable posture, where equilibrium control is minimized and thus the APA may be more related to the task performance.

While sitting, the base of support is substantially larger and the center of mass is positioned closer to the base of support than in the standing position. Thus, the task of maintaining the center of mass projection within boundaries of the base of support is less challenging and one may expect different patterns of APAs before arm pointing.

The majority of studies on the APAs have been carried out in standing adults performing various types of arm movements (*Cholewicki, Polzhofer & Radebold, 2000*; *Sijper & Latash, 2000*; *Cecchi, Došlă & Marini, 2001*; *Juras & Słomka, 2013*) and the few existing reports while sitting are somewhat incomplete and conflicting (*Van Der Fits et al., 1998*; *Aruin & Shiratori, 2003*; *Van der Heide et al., 2003*; *Le Bozec & Bouisset, 2004*; *Cuisinier, Olivier & Nougier, 2007*). In particular, APAs are sometime absent in individuals performing reaching tasks while sitting (*Moore & Brunt, 1991*; *Van Der Fits et al., 1998*), while other studies, where seated subjects exerted maximal force on a bar, showed APAs in trunk and hip muscles (*Teyssedre et al., 2000*; *Le Bozec & Bouisset, 2004*). Although these studies have achieved interesting results, some of them did not involve lower limb muscle investigation, which are mainly responsible for ankle joint mobility (i.e., tibialis anterior and soleus). Indeed, a recent review of the literature showed that the effect a manipulated initial position, as the amount of feet and/or back support, or leaning toward a side, for example, was not studied (*Chikh et al., 2016*).

*Aruin & Shiratori (2003)* found that APAs in sitting with feet support compared to standing were attenuated in the leg muscles (tibialis anterior, soleus, rectus femoris, and biceps femoris) but not in trunk muscles (erector spinae and rectus abdominis). However, they only compared the amount of muscle activation and not the latency within a temporal window of 100 ms before the movement. Thus, they precluded conclusions on the activation order within the direction specific response. In addition, they did not worry about important aspects of the sitting posture, such as the amount of feet and/or back
support. Indeed, the quantity of reaction force exerted by the feet determines the center of pressure position along the antero-posterior axis and the torque for the task.

Accordingly, the present study aimed to disentangle the conflicting role of APAs during an upper-limb motion, using different postural stabilities (standing and sitting). For this, we used an arm-pointing task in which the subjects reach a central diode that suddenly turns on, while their postural stability was manipulated. In order to characterize the movement performance, we measured the kinematic parameter and registered the latency of lower limb muscles. We hypothesized that different arm kinematics features reflecting postural stability (i.e., arm shorter reaction time (*Papaxanthis, Pozzo & Schieppati, 2003*; *Berret et al., 2009*), arm velocity profile (*Stapley, Pozzo & Grishin, 1998*; *Stapley et al., 1999*)) will be present during unstable postures, suggesting different motor plans for the execution and the control of arm movements performed in different equilibrium context. We further hypothesized that whether the main concern of APA is the compensation of postural perturbations (i.e., consequences of COM displacement), then the increase of body stability should induce attenuation or a modification of APA patterns. On the other hand, if APA serves the preparation for forthcoming upper limb movement (i.e., in order to accelerate the center of mass) to perform the task accurately, one could expect the presence of similar APAs patterns independently of the equilibrium constraint.

# EXPERIMENTAL PROCEDURES

## Subjects

Ten healthy men between the ages of 21 and 28 years (mean height $1.71 \pm 0.04$ m, weight $71.8 \pm 6.2$ kg), with normal or corrected to normal vision, no known neurological or muscle disorders, took part in the study. All subjects were right-handed. They gave their informed consent. The study was approved by the Federal University of Pará's ethics committee (Ethical Application 46943215.0.0000.0018), before the beginning of the study.

## Experimental set up and protocol

Subjects either stood on the floor barefoot (Up) or were seated on the edge of a height-adjustable chair with (Sit Sup) or without feet on a force platform (SitUnsup). In all seated postures, we preconized that the surface in contact with the chair was 30% of the thigh length (i.e., from the head to the femur to the intra-articular line of the knee). To ensure any contact of the feet with the ground we adjusted the height of the chair, in the SitUnsup posture. In the SitSup posture, we standardized the ground reaction force (Fz) not to exceed 5% of the total Fz during stance (Fig. 1).

They were required to keep their eyes fixed on a horizontal bar, placed in front of them at 2 m from the floor and 2.5 m from the participants' feet, with a central diode aligned with their right shoulders. Participants were asked to point with their index finger towards the central diode, which was suddenly turned on. Subjects stayed with the left arm down along the body and the right index finger pointing towards the ground, with an angle of between 30° and 35° between the arm and trunk. In all trials, participants were told to raise their arm as fast as possible and to start as quickly as possible after the appearance of the visual stimuli, remaining for a few seconds with their arm in the air, and to move their

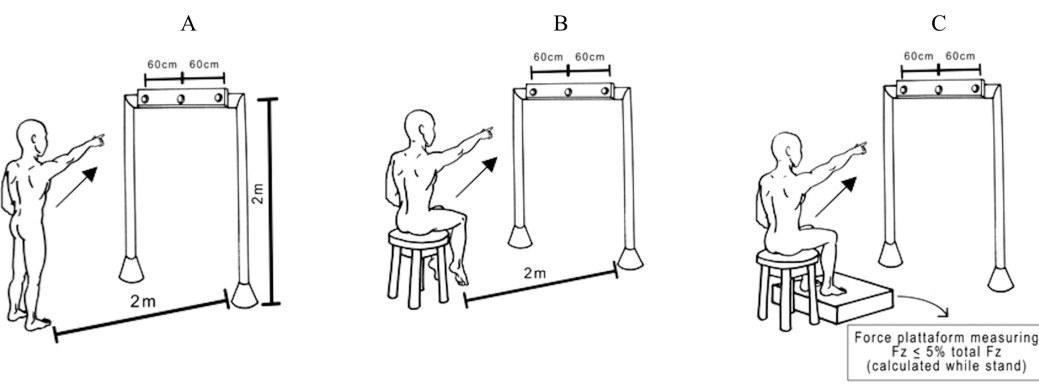

**Figure 1  Experimental pointing task.** View of the experimental set-up for the task showing a participant in the final posture and the three possible targets. The central diode of the bar between the two laterals was situated exactly in front of the participants' right shoulder. Participants were asked to point their index finger at the central diode which suddenly turned on. Postures: (A) Up; (B) SitUnsup; (C) SitSup.

index finger back towards the initial starting position. Randomized blocks of ten trials in each posture were performed with a 5 min rest between them.

## Kinematic and electromyographic recording

A three-dimensional motion analysis system (Simi Motion), with two cameras at a sampling frequency of 120 Hz, was used in order to record the participants' movements. The participant had eight infrared reflective markers placed at the main joints of right upper limb (i.e., index; wrist; elbow and shoulder).

Surface electromyographic (EMG) data, from the dominant-side leg and trunk muscles using disposable self-adhesive electrodes (Red Dot, 3 M): tibialis anterior (TA), soleus (SOL), rectus femoris (RF), semitendinosus (ST) and Deltoidis anterior, were recorded using two EMG devices (Emgsys 30306®; EMG System do Brazil, São José dos Campos, Brazil), with a sampling rate of 2 kHz and a frequency spectrum of 20–500 Hz. The EMG signals were amplified (4,000), and digitized with a 16-bit resolution. The participants' skin was prepared for Ag/AgCl electrodes (Med-Trace 200; Covidien Kendall, Dublin, Ireland) using Nuprep® (Weaver and Company— Aurora, United States) and alcohol. The active electrodes were placed on the muscles at 20-mm intervals, and the reference electrode was placed on the spinous process of the seventh cervical vertebra based on the orientations proposed by the Surface Electromyography for Non Invasive Assessment of Muscles guidelines (*Hermens et al., 2000*).

## Data analysis parameters and statistics
### Kinematic data

The kinematic parameters of each marker (trajectory and tangential velocity profile) were analyzed using MATLAB (MathWorks, Natwick, MA, USA). The data on $x$, $y$ and $z$ axis were filtered with a 10-Hz low-pass, second order Butterworth filter. We defined the total movement duration (MD) as the time interval of the index tangential velocity profile that exceeded five percent of its maximum value. Moreover, we calculated the movement

velocity (MV) and the reaction time (RT). Finally, we also examined the differences of index tangential velocity profile in function of participant postures. For this, we calculated the ratio of acceleration time to total movement duration (i.e., time to peak velocity).

### EMG data

The data were synchronized and analyzed off-line with MATLAB programs. All signals were filtered with a 100-Hz low-pass, second-order Butterworth filter, and EMG signals were rectified. Individual trials were viewed off-line on a monitor screen.

After additional visual inspection of the data, Tzero moment, as defined by the marker on the finger, was considered as the instant when the tangential velocity of the marker reached 5% of PV during that particular trial (*Bertucco & Cesari, 2010*). After alignment, trials within each series were averaged for each subject. To quantify the anticipatory changes in the muscle activity prior to movement, EMG signals were integrated from $-150$ with respect to Tzero ($\int$EMG150). This was further corrected for background activity, defined as the integral from $-500$ to 450 ms with respect to Tzero ($\int$EMG50) as follows:

$$\int EMG = \int EMG\,150 - 3 * \int EMG\,50.$$

The muscle latency was detected in a time window from $-450$ ms to $+200$ ms in relation to Tzero by a combination of computer algorithm and visual inspection of the averaged trials. The latency for a specific muscle was defined as the instant lasting for at least 50 ms when its EMG amplitude was greater (activation) or smaller (deactivation) than the mean of its baseline value, measured from $-500$ to $-450$ ms, plus 2 SD (*Aruin & Shiratori, 2003*).

## STATISTICS

Statistical procedures were performed in RStudio (R version 3.3.2, *R Core Team (2016)* and RStudio 1.0.136, *RStudio Team (2016)*) and included repeated-measures ANOVA with body posture (sitting with support, sitting without support and standing) as factor. Post-hoc analyses were done with Tukey HSD tests when necessary. For all these statistical treatments, the significance level was set at $p < 0.05$.

## RESULTS

### Kinematic characteristics

Kinematic' characteristics are summarized in Table 1. Finger displacement showed that a similar trajectory length was performed in the three posture [$F(2, 28.9)$, $p = 0.77$]. RT presented a main posture effect between the three postures [$F(2, 6.41)$, $p = 0.005$]. Post-hoc test revealed that RT was higher in the SitUnsup posture compared to the standing and SitSup posture ($p < 0.001$). MD also showed a main posture effect [$F(2, 3.56) = 0.0042$] and the post-hoc test showed the highest MD in the SitUnsup posture, compared to the others ($p < 0.001$). No statistical differences were observed for MV of the index movement between the three postures [$F(2, 2.46) = 0.104$]. The results also revealed a postural effect onto index tangential velocity profile [$F(2, 10.86) = 0.0003$]. In SitSup posture subjects

**Table 1  Comparison between Kinematic parameters.** Mean values kinematic parameters. Values are given as mean (SD).

| | Finger displacement (cm) | Reaction time (ms) | Movement duration (ms) | Velocity (m/s) | Acceleration time/movement duration |
|---|---|---|---|---|---|
| SitSup | 118.74(6.13) | 407.31(21.03) | 408.12(68.72) | 0.48(0.03) | 0.46(0.06)[**] |
| SitUnsup | 116.90(8.22) | 433.71(13.07)[*] | 529.06(127.71)[*] | 0.46(0.03) | 0.38(0.04) |
| Up | 116.48(7.83) | 411.01(18.50) | 444.56(106.64) | 0.45(0.02) | 0.37(0.03) |

**Notes.**
[*] $p < 0.005$ difference between SitUnsup and the two other postures.
[**] $p < 0.005$ difference between SitSup and the two other postures.

had higher ratio of acceleration time to total movement duration comparing to the Up and SitUnSup postures.

This indicates that the temporal parameters of the upper-limb movements were affected by the postural conditions whereas their spatial features remained similar.

## Muscle activation timing between postures

Figure 2 shows the rectified EMG data obtained for one trial for a typical subject comparing the tree postures.

Figure 3 shows the scatter of the overall dataset of the timing of muscle activations (all the trials of all participants in the three protocols).

In the Upright posture, muscles participated in the APAs in the following order: the proximal lower limb muscles (ST: 110 ms ± 10 and RF 80 ms ± 20) followed by the distal lower limb muscles (SOL: 80 ms ± 10 and TA: 70 ms ± 30). While seated this order were reverse: TA and SOL had earlier onset [SitSup (TA: 90 ms ± 20 and SOL: 80 ms ± 20) and SitUnsup (TA: 90 ms ± 10 and SOL: 100 ms ± 10], followed by ST and RF [SitSup (ST: 90 ms ± 20 and RF: 80 ms ± 10) and SitUnsup (ST: 90 ms ± 10 and RF: 80 ms ± 10)].

Comparing the timing of muscle activation between the postures, the ANOVA showed a main posture effect for TA [$F(2, 591) = 18.18$, $p < 0,000001$] and ST [$F(2, 591) = 18.18$, $p < 0,000001$]. The post-hoc analysis revealed earlier activations of the ST in Upright posture ($p < 0.001$) compared to both seated posture and earlier onset of TA in seated posture ($p < 0.001$) compared to Up.

## Muscle activation rates and magnitude between postures

To demonstrate the consistency of the results, we calculated the activation rate for each muscle in each posture, which corresponded to the percentage of trials showing significant muscle activation (burst). This corresponded to a minimal of 60% of trials with burst.

The normalized integrated electromyographic activity (EMGi) are summarized in Table 2. SOL and ST showed a similar behavior and presented a main posture effect between the three postures [$F(2, 6.31)$, $p = 0.003$]. Post-hoc test revealed that both SOL and ST had higher APA integral in Up posture compared to SitUnsup and SitSup ($p < 0.001$). RF also showed a main posture effect [$F(2, 4.56) = 0.002$] and the post-hoc test showed the lowest APA integral in the SitUnsup posture, compared to the others ($p < 0.001$). The results also revealed a postural effect onto TA APA integral [$F(2, 9.84) =$

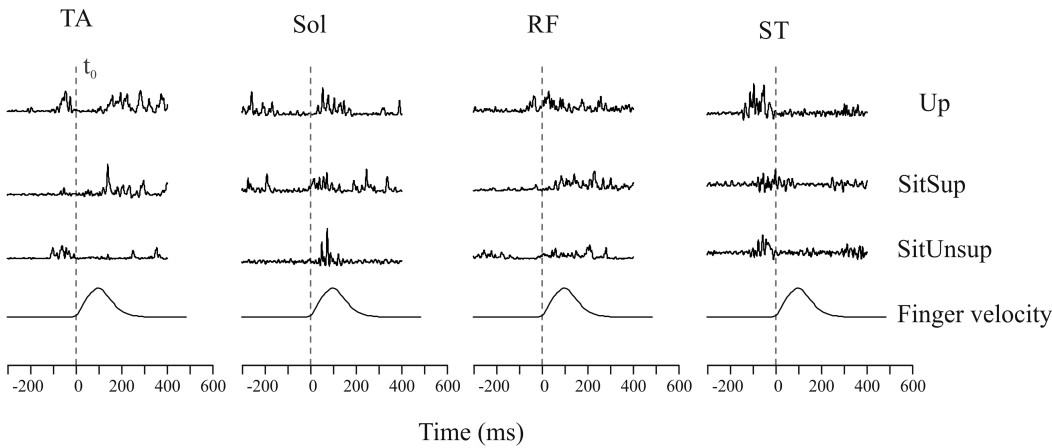

**Figure 2 Raw muscle activity of a typical subject recorded during one single trial.** Plotted signals were just rectified and normalized with respect to their maximum values recorded over all trials. The vertical dashed line mark finger movement onset (**tzero**). The time interval between 300 ms before movement on-set was considered. Up, upright posture; SitSup, Sit posture with contact feet support; SitUnsup, sit un-supported posture; Muscles, DEL, Deltoideous; ST, Semitendinosus; RF, Rectus Femoris; SOL, Soleous; TA, Tibialis Anterior.

**Table 2 Comparison between normalized integrated electromyographic activity (EMGi).** Normalized integrated electromyographic activity (EMGi) of muscles Mean values integral parameters (%). Values are given as mean (SD).

|  | TA (%) | SOL (%) | RF (%) | ST (%) |
|---|---|---|---|---|
| Sit Sup | 67.39(27.30)** | 52.11(28.12) | 45.07(20.85) | 40.08(28.21) |
| Sit Unsup | 45.37(35.02) | 50.87(30.63) | 24.40(20.20)*** | 40.43(28.63) |
| Up | 49.52(32.04) | 75.59(15.26)* | 57.27(27.73) | 55.84(23.92)* |

Notes.
*$p < 0.005$ difference between Up and the two other postures.
**$p < 0.005$ difference between Sit Sup and the two other postures.
***$p < 0.005$ difference between Sit Unsup and the two other postures.

0.0003]. In SitSup posture subjects had higher values comparing to both Up and SitUnsup postures.

## Temporal organization of finger and knee displacements

After processing the EMG data and observing the movement execution we decided to repeat the experiment with four additional subjects. Indeed, the visual observation of the pointing task performed with a SitUnsup posture showed a clear posterior displacement of the ankle joint (i.e., knee flexion). To quantify such displacement and its relation with the upper-limb motion we recorded the kinematic of three additional markers placed on the right lower limb (i.e., 5th metatarsal; ankle; knee and hip).

Pointing task movements demonstrated linear displacement of the ankle dependent of the adopted posture. Figure 4 illustrates the forward/backward trajectory this joint related to the upward finger displacement. We observed that while seated without feet support, subjects adopted a synchronous flexion of the knee (i.e., backward displacement of the

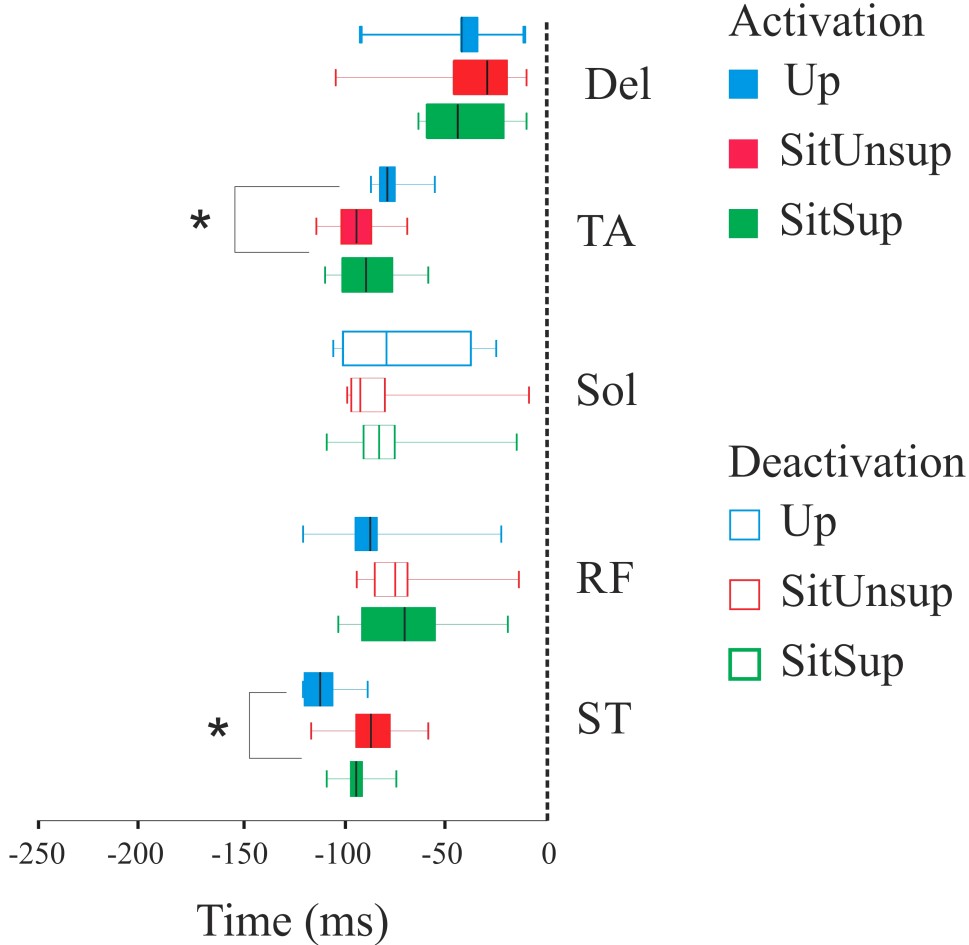

**Figure 3** **Latency (activation/deactivation before tzero) for lower limb muscles.** Up, upright posture; SitSup, sit posture with contact feet support; SitUnsup, sit unsupported posture; Muscles, DEL, Deltoideous; ST, semitendinosus; RF, Rectus Femoris; SOL, Soleous; TA, Tibialis Anterior. *$p < 0,05$ two-way ANOVA and post hoc test (differences between SIT SUP × UP and SIT UNSUP × UP).

ankle) with the finger displacement. Indeed, a backward displacement of the ankle began 30 ms ± 1.0 before the finger upward displacement. This pattern did not happen in the other two postures and suggests a synergic pattern of lower-upper limbs, accompanying arm movement.

## DISCUSSION

The present study aimed to verify whether and how APAs patterns and kinematic features changes while the subject postural stability was manipulated. Indeed, two seated postures (with and without feet support) and a standing posture were adopted while subjects performed an arm pointing task. These different postures permit to modify the degrees of postural stability. Accordingly, three postures were used, from high (i.e., sitting with feet support) to low stability (i.e., standing), as well as an intermediate position with a larger

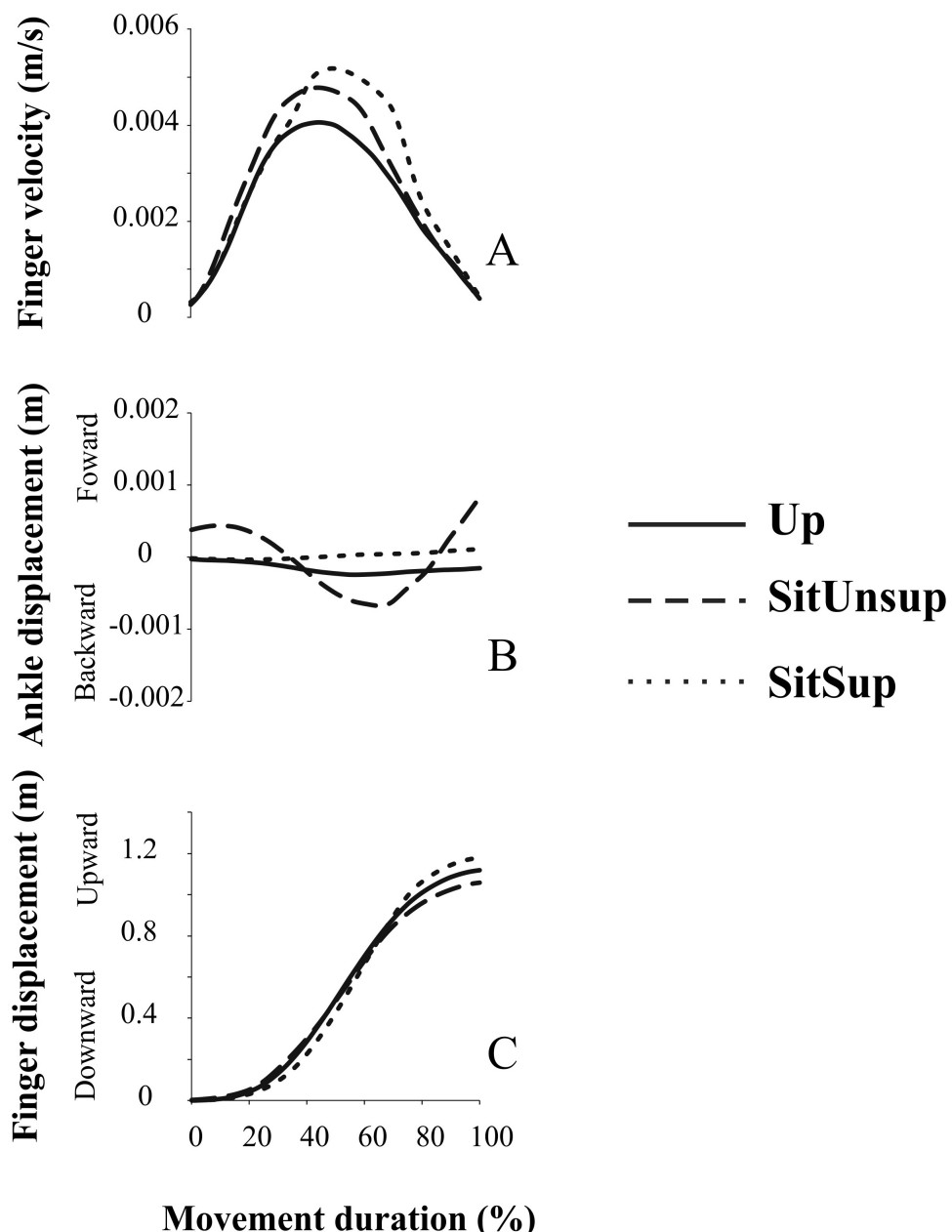

**Figure 4  Kinematic synergy a typical subject recorded during one single trial.** Plotted signals were normalized with respect to their total movement duration. Up, upright posture; SitSup, Sit posture with contact feet support; SitUnsup, sit unsupported posture. (A) Finger velocity; (B) ankle displacement; (C) finger displacement.

base of support but without feet contact that generated a higher instability than in sitting with support. The main results revealed that kinematic parameters and APAs latencies are dependent of the postural chains/configuration as well as the degree of stability that these last ones promote.

## Kinematic features

Different kinematic patterns were expected due to the postural stability manipulation since the base of support was limited to the feet in the Up posture and defined by the buttocks in SitUnsup or both buttocks and feet in the SitSup posture. As previously predicted, challenging postural stability induced different planning evidenced by an increase of the RT and MD for the seated condition without support compared to seated with support or standing. This suggests that arm motor plan becomes more challenging in the SitUnsup posture as indicated by the greater RT that is classically interpreted as an index of task difficulty (*Cuisinier, Olivier & Nougier, 2007*). One possibility would be that sitting without support does not give efficient mechanical mean (i.e., buttock) to accurately control posture and movement, thus challenging the motor planning. This is partially verified when considering the arm velocity profile which was significantly affected by postural stability. In agreement with our initial prediction we found greater max velocity and shorter deceleration duration of the arm pointing movement in the most stable position. Indeed, the velocity profile (or ACC/MD ratio) is classically described as reflecting the content of motor planning (*Abend, Bizzi & Morasso, 1982*; *Papaxanthis, Dubost & Pozzo, 2003*). More specifically, it is known that movements performed in the vertical plane used similar motor planning independently of the effector, and are directly dependent of the mechanical effect of gravity force field that change during upward and downward movement (*Papaxanthis, Dubost & Pozzo, 2003*). Herein, our result suggests that the equilibrium constraints rather than postural configuration were taken into account to plan an upper arm movement. This confirms the greater difficulty to plan the task (i.e., smaller pic of velocity) when a subtask (equilibrium) is added to the pointing one. Another point is that, although the APA was similar between the two-seated posture (see APA features bellow), the acceleration time and movement duration ratio (ACC/MD) was higher in the SitSup posture. If the role of APA was only to counteract the perturbation, we would expect to find a different behavior of temporal activation between theses postures. Our results also extent a recent research form *Stamenkovic & Stapley (2016)* which demonstrated that the muscle spatial recruitment was in favour of assisting initiation of movement in reaching during stance.

Since subjects were free to modulate the kinematic features of the movement, one could expect that the CNS would only use APA to minimize the perturbing effect of the dynamics of the movement, as described in the classical paradigm. However, we did not observe differences in the velocity of the movement between the positions in opposite of previous papers. This means that the perturbation was equal, although APA magnitude, as it will be described in the next topic, were different in the upright position, compared to the other two. Previously, it was described that APA magnitude is scaled with movement velocity (*Bertucco & Cesari, 2010*), which was not the case in our paper. This strongly suggests that APA has an additional contribution than the classical paradigm.

## APA features

Concerning the EMG latencies of the lower limb muscles (ST, RF, SOL and TA), we observed the presence of APAs independently of the posture, confirming our prediction. Stable postures may not require an APA stabilizer, but the presence of APAs regardless

of the stability suggests an additional role for APA beyond the feedforward control of the other body parts (i.e., as accelerators, or to facilitate the pointing movement). Indeed, for all postures, the activity of these muscles anticipated the arm pointing movement replicating the classical deactivation of anterior muscles (RF and TA) and activation of posterior muscles (SOL and ST) (*Teyssedre et al., 2000*; *Chiovetto, Berret & Pozzo, 2010*). By showing the presence of APA whatever the stability condition, our results demonstrated that adults performing arm movements while seated also present direction-specific activity preceding the focal muscle. This supports the idea that APAs may be used either to facilitate the task and/or to control the postural instability by activation of the same trunk muscles in different feedforward patterns (*Aruin & Shiratori, 2003*) according to the postural configuration (i.e., sitting vs. standing posture).

Another point is that, although the APA was similar between the two seated posture, the acceleration time and movement duration ratio (ACC/MD) was higher in the SitSup posture, which indicates a greater facility to plan the movement in this situation. If the role of APA was only counteract the perturbation, we would expect to find a different behavior of temporal activation between theses postures.

However, even if we observed APAs for all postures, the temporal pattern of these APAs differed and was dependent of the postural chains/configuration. While the thigh muscles (ST and RF) were activated earlier than the lower leg muscles (SOL and TA) in Up posture, the order was inverted when seated (both, with or without support). Therefore, our results demonstrated that sitting posture, regardless the stability, was associated with APAs in a different pattern compared to standing.

In standing posture the earlier events are located on the proximal muscle RF and ST while in sitting this is the distal one (TA). Anticipation of ST in the Up posture appears to be related to its role in the hip extension in order to counteract the forces on the pelvis produced by the reaching task and producing a trunk forward tilting movement (*Hodges, Cresswell & Thorstensson, 1999*; *Pozzo, Ouamer & Gentil, 2001*). Indeed, the seated subjects had the pelvis stabilized by the natural postural chain (*Forssberg & Hirschfeld, 1994*; *Van Der Fits et al., 1998*; *Le Bozec & Bouisset, 2004*; *Cuisinier, Olivier & Nougier, 2007*; *Vette et al., 2010*) and ST acted as knee flexor instead of hip flexor and also activated after the lower leg muscles. This is supported by our kinematic data about the SitUnsup posture that highlighted a simultaneous upward finger (arm) and backward ankle displacement. We propose that this kinematic profile reflects a strategy dedicated to facilitating the upper-limb movement in this posture. Two assumptions may be related to that. First, the initial muscle tone is different from standing to sitting postures (i.e., different joint positions, muscle length) and thus the following activity of these muscles change with respect to the resting state of the muscles. Second, the upward acceleration of the arm produces a forward trunk tilting that can be counteracted when sitting with support but is more difficult when no support is added on the feet. Therefore, APAs not only serves the whole-body equilibrium to the movement initiation, but are also concerned with providing the correct postural set to obtain the correct movement (*Cavallari et al., 2016*).

The APA magnitude of SOL and ST are in line with the literature that states that more APA is necessary to counterbalance the disturbances under more challenge stability

situations. For RF muscle, the SitUnsup posture showed even lower magnitude compared to the other postures, and this may be related to the ankle displacement observed in this position (i.e., activation of the antagonist ST). Surprisingly, TA did not show this behavior, and had higher magnitude in the SitSup position. This refutes the reasonable that APA magnitude is strictly related to the mechanical perturbation. Although we did not record the center of pressure (COP) data, perhaps because this muscle had some participation in COP displacement, accompanying the movement, in forward direction. This may not have happened in the SitUnsup due to the kinematic pattern imposed by lack of feet contact and the ankle backward displacement.

Contrarily to the SitUnsup posture, the SitSup posture did not present knee displacement due to the constraint of feet contact with the floor. However, the EMG pattern between both sitting posture was the same, since the postural chain was not modified. We also found that the TA activity appeared earlier for the seated postures compared to standing posture, acting as an ankle dorsiflexor (*Erim et al., 1996*; *Aruin & Shiratori, 2003*; *Van der Heide et al., 2003*). This early activity of TA in standing posture did not happen since the mechanical load in this joint is totally different and the body weight is completely loaded over it, delaying its participation. The fact that we observed different APA patterns between the sitting postures and the standing condition and not between the stability levels confirm our predictions (see Table S1). Thus, our results strongly suggest that APA acts to facilitate the limb movement and to counteract perturbation forces. Kinematics features, on the other hand, seem to reflect less challenging task and simple motor plan when the body is stabilized.

More studies in the field are required to add new perspective to the field of rehabilitation, for assessment and treatment.

## CONCLUSION

The present study showed that the APAs seem to be related to the stability conditions since we observed differences of EMG features for the two seated postures whereas the equilibrium constraints differed. Conversely, the degree of stability seems particularly involved in the motor planning of the upper-limb during a pointing task whereas the muscle postural chain (sitting vs. standing) was also determinant for APAs.

### Funding

This research was supported by the following grants: Pará Amazon Research Support Foundation (FAPESPA) grant number #180/2012; and Coordination for the Improvement of Higher Education Personnel (CAPES) / COFECUB research grant number #819-14. The funders had no role in study design, data collection and analysis, decision to publish, or preparation of the manuscript.

### Grant Disclosures

The following grant information was disclosed by the authors:

Pará Amazon Research Support Foundation (FAPESPA): #180/2012.
Coordination for the Improvement of Higher Education Personnel (CAPES) / COFECUB:
#819-14.

## Competing Interests

The authors declare there are no competing interests.

## Author Contributions

- Bianca Callegari conceived and designed the experiments, performed the experiments, analyzed the data, authored or reviewed drafts of the paper.
- Ghislain Saunier conceived and designed the experiments, authored or reviewed drafts of the paper.
- Manuela Brito Duarte and Gizele Cristina da Silva Almeida performed the experiments, analyzed the data, prepared figures and/or tables.
- Cesar Ferreira Amorim analyzed the data, contributed reagents/materials/analysis tools, authored or reviewed drafts of the paper.
- France Mourey and Thierry Pozzo conceived and designed the experiments, contributed reagents/materials/analysis tools, authored or reviewed drafts of the paper.
- Givago da Silva Souza performed the experiments, analyzed the data, contributed reagents/materials/analysis tools, prepared figures and/or tables, authored or reviewed drafts of the paper.

## Human Ethics

The following information was supplied relating to ethical approvals (i.e., approving body and any reference numbers):

The Federal University of Pará granted Ethical approval to carry out the study within its facilities (Ethical Application 46943215.0.0000.0018).

## Data Availability

The raw data is provided as Data S1.

## Supplemental Information

Supplemental information for this article can be found online at http://dx.doi.org/10.7717/peerj.4309#supplemental-information.

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
