# Peer review of "Anticipatory Postural Adjustments and kinematic arm features when postural stability is manipulated"

_PeerJ, doi:10.7717/peerj.4309_

## Round 0.1 · original submission · Major Revisions

Both reviewers thought the manuscript is interesting and provided several comments to clarify the text, methods and results.

Reviewer 1 ·

Basic reporting

The manuscript is very interesting and the meanly suggestion is to review the English because some words is not clear for manuscript language.

In the abstract is the first time that appears the acronym APAS should be written in full

Improve the description of the methodology because it is very succinct

Conclusion: not clear

Introduction is too long
Sample size calculation was done?
I suggest completing the information on the approval of the institution's ethics committee
The results are described in a very extensive way and with many values that I suggest that they come only in the tables and only those that are most relevant to the study are described.

What is the contribution of this study to clinical practice?

Experimental design

The methodology is well described and with details that allow to replicate the study

Validity of the findings

The findings is interesting for clinical practice

Additional comments

The manuscript is very interesting and the meanly suggestion is to review the English because some words is not clear for manuscript language.

Reviewer 2 ·

Basic reporting

The manuscript was written in professional english language.
There are few typos the authors should correct:
- Line 122: there is a double parenthesis in the references
- Line 152: "30% of the tight length" should be corrected to "30% of the thigh length"
- Line 216: I believe "man" is mean.

It is suggested to avoid abbreviations in the discussion, such as "mvt" line 350.

The authors should be consistent with the number styles according with journal requirements, decimal mark is sometime with point or comma.
There is an error to report APA timing: the values unit is in ms which is physiologically unlike for APAs (e.g. line 270, SOL 0,08 ms). Also, line 311 for ankle displacement, i.e. 1.3 ms.

The introduction and the references are relevant to introduce the background for the study

Figures and tables follow the conform to the journal requirements.

Experimental design

Since the biomechanical characteristics of the experimental tasks, the trunk muscles, such as erector spinae and the rectus abdominis, should have been recorded. They represent a critical group of muscles that would properly discriminate the APAs features among the 3 task conditions.
Why did not the authors record these muscles?

It is not clear how the acceleration time of kinematic data was calculated?

In the EMG data section is not described how the deactivation of muscle activity was calculated.

Please, indicate the software used for the statistical analysis.

The method to quantify the APA (in terms of integrated EMG signal) was specified in the methods. However, the results were not reported in the results sections

Validity of the findings

The aim of the study was to disentangle the conflicting role of APAs during a upper-limb motion using different postural stabilities (standing and sitting).
The results show different kinematic features among the 3 conditions mostly due to different postural constraints. Different task demands showed a different modulations of APA timing, which is in accordance to previous findings (e.g Bertucco & Cesari, 2010), and it supports in part the hypothesis of the authors of APAs role as postural strategies to facilitate the upcoming movement.

However, the 3 postural conditions influenced the kinematic performance in term of movement time and acceleration time. I am wondering whether the self-selected motor execution imposed by the subjects may have influenced, as a consequence, the APA magnitude and timing parameters, which in part would support the classical paradigm of APA as strategies to counteract the expected mechanical effect of the focal perturbation. In other words, since the subjects modulated the kinematic of the movement due to different postural constraints and difficulty, the APA therefore were differently modulated by the CNS to minimize the perturbing affect of the dynamics of the movement. The authors should reason this in the discussion.

While the quantification the APA (in terms of integrated EMG signal) was specified in the methods, this is completely missed in the results sections. The magnitude of the APA is a very critical feature that explains how the CNS generates these feedforward strategies to respond to the postural demands. This would better clarify, together with timing, the role of APA during the movement preparation. I anticipate that while APA timing reflects more the difficulty of the task and provide the underlying mechanics to facilitate the movement, on the other hands the APA magnitude would preparare the mechanical basic to counterbalance the disturbances arising from the volitional movement.

---

## Round 0.2 · Minor Revisions

The manuscript can be accepted after correcting the typo identified by reviewer 2.

Reviewer 1 ·

Basic reporting

The authors have made all corrections suggested by the reviewers and the article may be accepted for publication.

Experimental design

Suitable

Validity of the findings

The findings are interesting

Additional comments

The authors have made all corrections suggested by the reviewers and the article may be accepted for publication.

Reviewer 2 ·

Basic reporting

The Authors properly responded to my comments

minor comments:
line 114: typo, it should be thigh length NOT thight length

Experimental design

The Authors properly responded to my comments

Validity of the findings

The Authors properly responded to my comments

---

## Round 0.3 · accepted · Accept

The authors have corrected the typo identified by the reviewer.